# Metabolomics in Team-Sport Athletes: Current Knowledge, Challenges, and Future Perspectives

**DOI:** 10.3390/proteomes10030027

**Published:** 2022-08-10

**Authors:** Tindaro Bongiovanni, Mathieu Lacome, Vassilios Fanos, Giulia Martera, Erika Cione, Roberto Cannataro

**Affiliations:** 1Performance and Analytics Department, Parma Calcio 1913, 43122 Parma, Italy; 2Department of Biomedical Sciences for Health, Università degli Studi di Milano, 20129 Milano, Italy; 3Laboratory Sport, Expertise and Performance (EA 7370), French Institute of Sport (INSEP), 75012 Paris, France; 4Neonatal Intensive Care Unit, AOU Cagliari Department of Surgery, University of Cagliari, 09124 Cagliari, Italy; 5Department of Nutrition, Spezia Calcio, 19136 La Spezia, Italy; 6Department of Pharmacy, Health and Nutritional Sciences, University of Calabria, 87036 Rende (CS), Italy; 7GalaScreen Laboratories, Department of Pharmacy Health and Nutritional Sciences, 87036 Rende (CS), Italy

**Keywords:** metabolomics, physical exercise, sportomics, metabolites, miRNA

## Abstract

Metabolomics is a promising tool for studying exercise physiology and exercise-associated metabolism. It has recently been defined with the term “sportomics” due to metabolomics’ capability to characterize several metabolites in several biological samples simultaneously. This narrative review on exercise metabolomics provides an initial and brief overview of the different metabolomics technologies, sample collection, and further processing steps employed for sport. It also discusses the data analysis and its biological interpretation. Thus, we do not cover sample collection, preparation, and analysis paragraphs in detail here but outline a general outlook to help the reader to understand the metabolomics studies conducted in team-sports athletes, alongside endeavoring to recognize existing or emergent trends and deal with upcoming directions in the field of exercise metabolomics in a team-sports setting.

## 1. Introduction

Metabolomics is defined in the Oxford English Dictionary as “the scientific study of the set of metabolites present within an organism, cell, or tissue” [1]. In 2001, metabolomics was well-defined by Fiehn as the comprehensive and quantitative analysis of all metabolites of the biological system under study [2]. Later, Turnbaugh and Gordon described metabolomics as characterization via mass spectrometry, NMR, or other analytical methods of metabolites generated by one or more organisms in a given physiological and environmental context [3]. Thus, metabolomics is a discipline that studies the metabolome, which is the collection of small-molecule chemical entities (with molecular weights less than or equal to 1500 Daltons). In other words, this is the study of end-products of intricate biochemical pathways (deriving from genome, transcriptome, and proteome metabolism) that occur within and outside the cell [4]. Thus, the characterization of the metabolome, the “secret language” by which cells communicate, can provide helpful contributions to the understanding of the complex interaction between genes and the environment. Thus, networks of specific biological processes or physiological phenomena are activated upon a given stimulus/perturbation, such as disease, pharmaceutical drug, environment, diet, or physical activity. Metabolomics is increasingly used in different research areas, from medicine to toxicology, from nutritional science to plant science, and from cells to other organs [5]. Metabolomics is also a promising tool for studying exercise physiology and exercise-associated metabolism. This application in sports was recently defined with the term “sportomics” [6]. In the last decade, exercise metabolomics studies were conducted on runners [7], cyclists [8], soccer players [9,10], basketball players [11], and rugby players [12]. Indeed, exercise physiologists know well that exercise primes one for significant, often profound variations in the metabolism of numerous tissues and organs. The knowledge on exercise-induced variations in metabolites and the metabolic pathway is exciting and challenging [13,14]. To date, from 114,100 to 217,920 different metabolites have been observed in the human body, and more than 46,000 metabolite or metabolic signaling pathways have been described [15]. However, the majority of the published exercise/physiology metabolism studies measured fewer than dozens of metabolites and examined only one to two pathways at a time [16]; this scenario limits our knowledge of the complex connections between exercise, physiology, and metabolism. To address those limitations, over the past 20 years, exercise physiologists have performed minimally invasive and non-invasive metabolomics studies to assess the phenotype and physiology of sport and exercise [17]. The growing popularity of metabolomics in sport is due to the ability of metabolomics to simultaneously characterize several metabolites in several biological samples (e.g., saliva, urine, and sweat) obtained in non-invasive ways to gain a molecular snapshot of the impact of exercise [18]; other reasons and thus possible applications could be: to evaluate the acute effects of hydration, nutritional strategies to manage oxidative stress, and inflammation and immune response [8]; to determine how training perturbations can impact the metabolome over long or short periods and explore their effects over a long period of time [9]. The ability of metabolomics to probe processes that occur in real time or over hours or even days could explain why exercise physiologists are increasingly using this science. In this review on exercise metabolomics, we intended to provide an initial and brief overview of the different metabolomics technologies, sample collection possibilities, and further processing steps employed in the sports context and discuss data analysis and their biological interpretation. Thus, we do not cover sample collection, preparation, and analysis paragraphs in detail here but outline a general outlook to help the reader to understand the metabolomics studies conducted in team-sports athletes, alongside endeavoring to identify existing or emerging trends and deal with upcoming directions in the field of metabolomics applied to exercise in a team-sports setting.

## 2. Metabolomics Methods

Metabolomics is one of the most recent omics sciences which allows the simultaneous qualitative and quantitative analysis of different metabolites present in biological samples. Through the use of various platforms based on mass spectrometry (MS) such as capillary electrophoresis–mass spectrometry (CE-MS), liquid chromatography–mass spectrometry (LC-MS) and/or ultra-performance liquid chromatography–mass spectrometry (UPLC-MS), gas chromatography–mass spectrometry (GC-MS), or nuclear magnetic resonance (NMR) spectroscopy [19,20,21]. It is well-recognized that each platform has its advantages and disadvantages in metabolomic studies. In general, the use of nuclear magnetic resonance spectroscopy guarantees a more precise identification and quantification of metabolites; on the other hand, the analysis costs are higher, and the sensitivity is lower than different approaches based on mass spectrometry. Today, mass spectrometry is the most common platform used, mostly in combination with gas or liquid chromatography. Generally, GC-MS presents fast acquisition speed, robust data acquisitions, good sensitivity, and allows for the convenient identification of analytes using commercial databases and software. However, sample preparation is laborious, and identifying new compounds is difficult. Moreover, liquid chromatography–mass spectrometry can detect a broader range of polar and non-polar metabolites and has higher sensitivity but shows a limited ability to identify new molecules concerning NMR or GC-MS [20]. In LC-MS, polar and non-polar metabolites are separated using HILIC and RP columns, respectively. GC-MS, on the other hand, can separate both polar and non-polar metabolites through one type of column. The detailed pros and cons of these platforms have been discussed in some reviews [21,22]. Overall, chemical analysis using different methods (CE-MS, LC-MS, UPLC-MS, GC-MS, and NMR), the subsequent data processing (e.g., peak detection, de-noising, etc.), data reduction, and statistical analysis represent clear challenges to accurately identifying metabolites in a biological sample, as well as the biological interpretation and contextualization of detected metabolites [21]. Albeit with some differences in chemical methods, many metabolomics studies follow a similar workflow, which is shown in Figure 1.

Based on the analytic platform used and depending on the type of biological sample collected, the number of metabolites analyzed all at once can vary from dozens to thousands. Considering the variety of existing platforms and the heterogeneity of samples, it is not astounding to read published papers where different metabolomics methods have been used. For example, “metabolic fingerprinting” is a metabolomic technique commonly used in whole-organism metabolomic studies as well as in cell culture studies. This method is employed to characterize all measurable analytes in a sample, for example cell cytoplasm, serum, and plasma with the successive classification of samples and the categorization of diversely displayed metabolites, which define the sample categories [23]. Among several strategies that aim to quantify cell metabolites to increase our understanding of the complex interactions between metabolites’ level and the interpretation of metabolic networks, “metabolic footprinting” represents a commonly used method in microbiology and biotechnology to characterize extracellular metabolites, providing important information for functional genomics and strain characterization [24]. In the metabolomics scenario, there is a further classification of methods, and the most used is “targeted” and “untargeted” metabolomics, respectively. The first focuses on the proof of identity and the quantification of a specific, pre-defined group or category of a limited number (tens to hundreds) of well-interpreted and biochemically described metabolites in a tissue, biofluid, or biological matrix, and it is commonly used in clinical or biomarker discovery trials [25], while the second approach, known as untargeted or hypothesis-generating, focuses on the unbiased identification of the maximum number of metabolites or metabolic features in a bio-fluid, tissue, or biological matrix and it is predominantly used in biological discovery or hypothesis-generation applications. In both cases, the employment of nuclear magnetic resonance and liquid or gas chromatography coupled to mass spectrometry are predominant [26]. Overall, it is possible to obtain the accurate quantitation of target metabolites with triple quadrupole mass spectrometry, whereas the data obtained from untargeted metabolomics can be only semi-quantitative with high-resolution mass spectrometry.

In the literature, some authors refer to “metabolic profiling” as a synonym of metabolomics; in reality, this term is reserved correctly for the measurement in biological systems of the complement of metabolites and their intermediates that mirrors the response to genetic modification and physiological, pathophysiological, and/or developmental stimuli [27]. Other than the above-discussed methods, it is noteworthy that “lipidomics” involves the comprehensive analysis of all lipids, fatty acids, and lipid-like molecules in a biological or environmental sample [28].

## 3. Sample Collection and Processing

Concerning the choice of the type of sample, it is essential to consider the invasiveness of the collection method (invasive such as serum versus less invasive or non-invasive such as saliva), the suitability of the sample for the research question asked, and the chemical method used [29]. Typically, in metabolomics experiments, samples or bio-fluids that soak or surround the tissue or organ of interest, such as serum, saliva, urine, and stool, are chosen [30]. It is noteworthy that different sample types present both advantages and disadvantages, which are briefly reported in Table 1. In addition to the type, the numerosity of samples used for the experimental trial appears to be another important theme to consider. In several metabolomics studies, an average of 35 controls and 35 cases are mandatory to obtain adequate data. Sometimes, if metabolite change is important, five to ten cases and controls could be adequate [31].

After collecting the biological samples, they must be further processed or extracted to obtain a sample appropriate for the chemical metabolomics analysis. This depends on the type of the analytical platform and on the biological sample being used. For example, cells, feces, and tissue must be frozen, then crushed in powder, and extracted with chloroform and water or methanol solvents to be processed using NMR spectrometers or mass spectrometers [32], while when an already-liquefied sample is collected, for example, serum, plasma, or urine, the bio-fluid must be filtered to remove cellular debris, macromolecules, and proteins. The removal of the latter is critical because it prevents enzymatic reactions, which may alter the synthesis of metabolites in the original sample. Otherwise, in some cases, even bio-fluid must be extracted with organic solvent when analyzed with LC-MS or GC-MS. Indeed, the use of organic solvents is the most “operative” method of all proteins and in extracting classes of metabolites. Methanol or methanol:water (1:1) can be used to extract polar metabolites from serum, plasma, and/or saliva, while chloroform mixed with methanol:water in a ratio of 2:2:1.8 can be used to extract non-polar metabolites from most bio-fluids or tissues and lipids. To use the GC-MS platform, metabolites need to react with specific chemical moieties to enhance their volatility or isotopically label them for enhanced liquid chromatography separation and mass spectrometry differentiation [15]. These processes ensure that samples are enzyme- and protein-free and avoid the presence of newly originated metabolites that were not in the initial sample [32,33].

## 4. Data Analysis and Biological Interpretation

In papers where a “targeted” approach is used, after carrying out spectral processing, the metabolites present in the biological sample are characterized. Afterward, the set of analytes is statistically processed with a final data reduction. In this case, almost all of the metabolites that are targeted in the study design are typically identified and quantified. As an alternative, in studies where the “non-targeted” approach is chosen, data reduction techniques are used before the metabolites are identified. These initial steps are mandatory and are planned to reduce thousands of spectral physical appearances into dozens of statistical characteristics that can be identified in metabolites [33]. In metabolomics studies, the use of statistical analysis is of fundamental importance, which guarantees an acceptable interpretation of the data, and generally, two main approaches are used: (i) unsupervised and (ii) supervised methods.

Similar to other omics sciences, metabolomics analysis generates large datasets that are often difficult to interpret. The most common unsupervised technique able to reduce the vastity of such a dataset, even increasing data interpretation and reducing information loss, is Principal Component Analysis (PCA), which detects the main “landscapes” and elucidates the variance of a dataset. PCA data are classically shown as a two- or three-dimensional plot. The plot classically shows several clusters of data points that share some grade of similar characteristics. A PCA loadings plot displays the features (metabolites or spectral bins) that are most strongly discriminating between clusters. In this way, PCA plots may be used to extract the most important differentiating spectral or metabolite features in a metabolomic study. In this way, scientists reduce hundreds or thousands of components or metabolites to a manageable number of data [15].

However, it is important to point out that, among the common methods used in metabolomics, there is also the discriminant analysis of partial least squares (PLS-DA) and its optimized form, PLS-DA orthogonal (OPLS-DA). These methods of classification require vigilant validation and further testing to determine that the sorting model is not oversized. Overall, the use of these specific statistic techniques allows one to summarize the information contained in large data tables by means of a smaller set of variables in order to observe trends, clusters, and outliers. On one hand, these powerful statistical modeling tools provide insights into separations between experimental groups based on high-dimensional spectral measurements from different platforms such as NMR, MS, or other analytical instrumentation. On the other hand, when used without validation, these tools may lead researchers to statistically unreliable conclusions. Although both targeted and non-targeted approaches use distinct steps to identify metabolites, they share similar practices for biological interpretation. To date, exercise physiologists and metabolomics researchers use bioinformatic tools for the enrichment of metabolite sets from metabolomics trials. Among these libraries, Kyoto Encyclopedia of Genes and Genomes (KEGG) pathways [34], Reactome [35], MetaCyc Suite [36], and PathBank [37] are used alone or in combination. MetaboAnalyst is an additional regularly used web source that provides a large number of tools for more sophisticated biological interpretation, biomarker analysis, and multi-omics integration [38,39,40]. Another problem faced by analysts when interpreting the results of metabolite analyzes is linked to the type of detector used. For example, a time-of-flight (TOF) detector is a particle detector that can discriminate between lighter and heavier elementary particles. Gas chromatography coupled with mass spectrometry, including time-of-flight mass spectrometry (GC-TOF MS), is usually applied to determine metabolites with thermal stability and volatility [40]. Meanwhile, mass spectrometry-based multiple reaction monitoring (MRM) assays show superior multiplexing detection capabilities, are highly reproducible, specific, and are a very sensitive technique for quantifying targeted protein/peptides [15]. In addition, MRM has been considered to be one of the most effective tools available for quantitative clinical proteomes [6].

## 5. Team-Sport Athlete Studies

The complex and combined nature of the human organism to exercise response makes the use of metabolomics a useful method to fill gaps in our current understanding of exercise-associated cellular responses [41]. With this in mind, exercise physiologists have recently started to apply the metabolomic approach to sports to investigate the response of cells, tissues, and organs to specific physical effort. Table 2 provides an overview of previously published studies investigating the use of metabolomics analysis in team-sport athletes. To the best of our knowledge, the first application of metabolomics in team sports was published in 2014 by Santone and coworkers [42]. In this early study, the authors examined, in 14 elite professional soccer players, the impact of the level 1 Yo-Yo intermittent recovery on salivary metabolites. Saliva samples were collected before and after the Yo-Yo performance test and analyzed using a proton [^1^H]-NMR platform with a subsequent PCA analysis. Among hundreds of metabolites detected, significant temporal changes in metabolite concentrations for the identification of the pre/post intermittent Yo-Yo test were identified as follows: urea, glucose, lactate, citrate, acetate, glycerol, glutamate, leucine, alanine, and lysine. In 2014, Ra and collaborators [43] produced the first large-scale sportomics study, in male soccer players (n = 122) participating in successive soccer matches (three) over a period of 3 days, with the aim to identify potential metabolites of fatigue. They choose saliva as a bio-fluid, and the analysis was conducted on the CE-MS platform. They found that the levels of the following amino acids, valine, isoleucine, tyrosine, leucine, tryptophan, and phenylalanine, were increased meaningfully in the athletes with fatigue compared to the non-fatigued ones; even 3-methyl histidine was increased. In addition, another two metabolites, glucose-1 and -6-phosphate, were also impaired. These metabolites were recommended as potential targeted saliva-detectable compounds of fatigue in soccer players [43].

A few years later, Barton et al. [44] used metabolomics in an elite athlete cohort of international-level rugby players and age-matched controls to study host-derived and microbial-derived metabolic profiles using urine and fecal samples. The samples were analyzed using a combination of multiplatform metabolic phenotyping such ^1^H-NMR, GC-MS, and hydrophilic interaction UPLC-MS, and then, a multivariate analysis based on OPLS-DA was used to compare urine and fecal samples from athletes and controls. Athletes showed higher urine levels of trimethylamine-N-oxide (TMAO), L-carnitine, dimethylglycine, O-acetyl carnitine, proline, betaine, creatine, acetoacetate, 3-hydroxy-isovaleric acid, L-valine, acetone, N-methyl nicotinate, N-methyl nicotinamide, phenylacetylglutamine (PAG) and 3-methylhistidine. Furthermore, higher levels of trimethylamine (TMA), short-chain fatty acids, lysine, and methylamine were found in the fecal samples of athletes compared to the age-matched control cohort. In addition, the urine metabolome of athletes presented lower concentrations of glycerate, allantoin, and succinate. Similarly, the fecal samples showed lower levels of glycine and tyrosine when compared with control subjects. The highest levels of short-chain fatty acids in feces, in particular propionate, were correlated to protein intake, while butyrate was shown to have a strong association with dietary fiber intake. This information supports previous insights into the beneficial influence of physical exercise and associated diet on end-products such as short-chain fatty acids, which are notably associated with numerous health benefits [45,46]. Moreover, the highest presence of TMAO and especially of 3-hydroxy-isovaleric acid has been demonstrated to have efficacy for inhibiting muscle wastage and reducing exercise-induced muscle damage [6,47], while the highest presence of metabolites such as creatine, 3-methylhistidine, and L-valine mirror increased muscle turnover [48]. The high concentration of phenylacetylglutamine (PAG) in the urine of rugby players is not surprising, because it is a derived metabolite from phenylalanine, which is notably elevated in lean subjects [49].

A series of studies were published by Al-Khelaifi’s group during the two-year period 2018–2019, in which the authors conducted non-targeted metabolomics-based mass spectroscopy combined with ultra-high-performance liquid chromatography to metabolomics profiling spare samples collected by doping control in athletes of different sports [50,51,52]. The first pilot study conducted by this group showed that serum samples from athletes involved in different sports exhibit distinct xenobiotic profiles that may reflect drug/supplement use, diet, and exposure to various chemicals. For example, soccer players showed higher levels of caffeic acid, quinate, and ectoine. Hippurate was also higher, as well as 4-vinyl-guaiacol sulfate and 2-furoyl glycine [53]. In a second pilot study, metabolomics was used to compare the blood metabolic profiles between moderate-and high-power and endurance elite athletes and to identify the potential metabolic pathways underlying these differences. Out of 743 analyzed metabolites, gamma-glutamyl amino acids were significantly reduced in both high-power and high-endurance athletes compared to their moderate counterparts, indicating the importance of the glutathione cycle and overall oxidative stress substrates. Team-sports athletes (e.g., soccer, basketball, and volleyball) exhibited increased levels of testosterone and progesterone and decreased levels of diacylglycerols and eicosanoids (fatty acids metabolism). In addition, high-power athletes had increased levels of phospholipids and xanthine metabolites compared to their moderate-power counterparts [53]. One year later, the same group applied metabolomics to compare the metabolic signature of elite athletes with low/moderate cardiovascular demand (e.g., rugby, baseball, and volleyball) versus higher demand (e.g., basketball and soccer). Out of over 750 metabolites detected, 112 were associated with cardiovascular demand, and 40 of them (e.g., adenine, creatine, glutamine, carnitine, arachidonic acid, plasmalogen, cortisol, leucine, valine, and isoleucine) were increased in soccer players and basketballers compared to their low/moderate-cardiovascular-demand counterparts, while 70 compounds (e.g., glutamate, beta-citryl-glutamate, gamma-glutamyl amino acids, 5-oxoproline, fatty acid-carnitines, and acylated carnitines) were increased in baseball, rugby and volleyball players, mirroring an enhanced anti-oxidative stress scavenging mechanism and the higher beta-oxidation of fatty acids for energy generation during exercise in this class of athletes with low cardiovascular demand [52]. Genetic predisposition to elite athletic performance was also studied by investigating genetically influenced metabolites discriminating elite athletes from non-elite athletes, with the aim to identify those associated with endurance sports [53]. Pitti et al. [54] found changes in the concentrations of 56 metabolites after an official soccer match, studying the saliva samples of 17 female professional players. The authors reported an increase of 40% of total salivary proteins in starting players that played the entire match and of 20% for those entered, while no significant change was observed in those that were substituted or that did not participate in the match. Metabolomics was also applied by Akazawa et al. [55] to examine the impact of the quality of sleep (QoS) on metabolites and cognitive function in female volleyball athletes over a period of one week. Among several saliva metabolites detected, specific compounds showed differences between athletes with better and lesser QoS; for example, 2-oxoglutaric acid, ornithine, citrulline, lysine, proline, tyrosine, and arginine levels were higher in the QoS group (reporting a better QoS) compared to the those reporting a lesser QoS. Paradoxically, ergogenic caffeine was found in higher levels in the first group, while urea and myo-inositol 3-phosphate were in higher concentrations in the lesser QoS group. These results pointed out that the quality of sleep could affect energy metabolism and amino acid during heavy exercise and that a metabolomics approach can be a useful tool to track changes and potentially identify athletes with sleep disturbances. Most recently, Pintus et al. [9] tracked the first-morning urine of elite soccer players during the pre-season period in three-time points. Authors studied 63 urine samples collected from 21 male soccer players during the 2nd, 6th, and 16th day of the pre-season period. Urine sampled at the beginning of this training period was higher in TMAO and dimethylamine (DMA) levels, showing an increased intake of dietary protein from players. Urine sampled on sixth day showed an elevated increase in 3-hydroxybutyrate, citrate, and hippurate, mirroring an increase in fatty acid metabolism and an increased intake of phenolic compounds and a host-gut microbial co-metabolism. Finally, at the end of this period of preparation, urine presented higher levels of guanido-acetic acid, involved in the urea cycle and suggested to be a biomarker of exhaustive exercise and fatigue. In recent times, metabolomics was adopted by O’Donovan et al. to study the impact of training load or type of exercise on fecal and urine metabolites, independently of dietary intake. The authors classified elite-level athletes based on decreasing/increasing dynamic components and increasing/decreasing static components. Interestingly, metabolites such as succinic acid, cis-aconitate, and lactate in urine samples and creatinine in feces were found to be significantly higher in athletes involved in increasing high dynamic and decreasing low static components (e.g., field hockey and rowing), indicating muscle turnover may have been greatest in this cohort of athletes [55].

Another interesting and useful application of metabolomics is offered by the study of Quintas et al. [56]. Data of 80 professional young soccer players were collected in a longitudinal observational trial, analyzed using ultra-performance liquid chromatography coupled to electrospray ionization quadrupole time-of-flight mass spectrometry, and assessed using partial least squares regression. Metabolites associated with external load were hypoxanthines, tyrosine, tryptophan, hormone metabolites (hydrocortisone, tetrahydrodeoxycortisol, dihydrotestosterone glucuronide, androsterone glucuronide, cortolone-3-glucuronide, testosterone glucuronide, and tetrahydroaldosterone-3-glucuronide), intermediates in phenylalanine metabolism, 4-pyridoxic acid, and catabolic products of vitamin B6 and riboflavin [57]. These markers mirror exercise-induced adaptations and increased physical activity [57,58]. In the recent past, Hudson et al. [59] characterized the metabolic perturbations caused by competitive rugby. The authors collected blood, urine, and saliva samples every morning throughout a competitive match week during the early part of the competitive season, and samples were analyzed using NMR. The most important changes were observed post match play and included metabolites such as citrate, lactate, and alanine, explaining the TCA cycle intermediaries, the conversion of pyruvate in lactate ensuring glycolysis can continue, and for alanine, that gluconeogenesis exists to meet the total match energy needs. Other discriminatory metabolites identified after the match were succinic acid, acetoacetate, and acetone, mirroring reduced fatty acid oxidation during the match. After the match, alanine, histidine, and tyrosine levels were reduced, while leucine, 2-hydroxyisocaproate, and 3-hydroxy-3-methylglutarate levels were increased, showing the protein breakdown and degradation of this branched-chain amino acid occurred during a game. Of note, two days’ post match, alanine increased significantly, and similarly, 3-methylhistidine and glycylproline, and 4-hydroxyproline, well-known markers of muscle and collagen damage, were increased [59]. Metabolomics was also applied to identify metabolites and metabolic pathways associated with leukocyte telomere length (LTL), a predictive marker of biological aging. Among 837 metabolites measured in serum samples of 126 young elite male soccer players, 67 showed significant associations with LTL; in particular, glycine-serine-threonine, benzoate, and lysophospholipids were elevated with longer LTL. In contrast, monoacylglycerols, sphingolipids, long-chain fatty acids, and polyunsaturated fatty acids were enriched with shorter telomers. The authors found that glutamine, N-acetyl glutamine, xanthine, beta-sitosterol, N2-acetyllysine, stearoyl-arachidonoyl-glycerol, N-acetylserine, and 3–7-dimethylurate were the metabolites that best predicted LTL in these groups of team-sport players [60].

A study involving Brazilian male soccer players showed variances in the metabolic profile just after and 20 h post matches. This indicates the potential of this kind of analysis to distinguish metabolic profiles, after exercise, in the recovery process. The authors documented that athletes with higher session ratings of perceived exertion (s-RPE) have more metabolic variations related to muscular damage and energy metabolism in comparison to athletes with a lower s-RPE. Metabolites that increased at the end of both matches were: TMA, DMA, creatine, and creatinine, which reflect muscular stress related to the nearness to the match. In addition, at the end of the matches, metabolites such as glycine, hippuric acid, l-serine, gallic acid, and betaine were increased. They were associated with cellular regeneration, antioxidant, and anti-inflammatory actions [61].

Khoramipour et al. (2020) characterized and compared the metabolic fluctuations between four 10 min quarters of a high-level basketball match. The authors reported increased salivary concentrations of lactate, pyruvate, succinic acid, citric acid, glucose, and hypoxanthine after quarters 1 and 3, demonstrating more reliance on anaerobic energy systems and increased levels of ATP turnover during these two quarters. In contrast, after quarters 2 and 4, saliva presented reduced levels of valine, leucine and increased concentrations of alanine, glycerol, acetoacetic acid, acetone, succinic acid, citric acid, acetate, and taurine, fat metabolism and gluconeogenesis, describing how the accumulated fatigue and reduction in high-intensity activities in the second and fourth quarters reduced the speed of energy production, and players utilized more aerobic energy [62].

## 6. Limitations of Previous Studies and Potential Improvements

As with all other “omics” technologies, sport metabolomics is continually growing and incessantly improving. Herein, we examined studies published until now, and it is clear that several of the previous exercise metabolomics investigations conducted in team-sports athletes lacked statistical consistency, for example, multivariate statistics were extensively used, they were under-powered, and small sample sizes were included, and false discovery rates were not corrected. These precautions are expected of most “omic” studies published today. In general, bearing in mind this analysis of the existing literature, we found a disturbing lack of metabolite quantification in many published exercise metabolomics studies. The lack of the absolute quantification of these studies makes comparisons almost impossible. The reason for this is that different laboratories, distinct study designs, and/or diverse platforms (GC–MS vs. NMR vs. LC-MS) were adopted to identify and quantify metabolites.

Undoubtedly, if metabolomics is to be used in exercise, moving beyond the research phase with regulated protocols, demanding and more standardized protocols must be adopted for metabolite identification and quantification, as are recognized in clinical settings in the metabolomics field. Albeit the design of the experimental study for several metabolomics sports trials has significantly improved over the past 10 years, and more advances are conceivable [63,64,65,66].

Supplemental progress in more standardized data collection, analysis, and reporting protocols would also help to improve the overall comparability and quality of exercise metabolomics studies. Furthermore, an important goal for exercise metabolomics studies will be the use and the combination of additional omics techniques such as genomics, transcriptomics, and proteomics in the study design. Therefore, to really study metabolomics, other “omics” techniques are required to enable a deep comprehension of the physiology and the biology involved, together with the understanding of the interplay of gene expression, proteins, and metabolites with the environment.

## 7. Conclusions and Future Perspectives

We believe that a multi-omics approach, including proteomics, genomics, transcriptomics, and metabolomics approaches, would allow sports coaches, exercise physiologists, and sport nutritionists to help and assist elite athletes more efficiently via optimized dietary and exercise prescriptions. In practice, we propose that more emphasis in exercise metabolomics needs to be placed on human studies with more focus on practical-oriented and real-world designs using non-invasive sample collection methods such as urine and saliva.

Knowing these preliminary applications, we can expect that the use of multi-omics in sports science will continue to grow in both elite sports performance and clinical exercise settings. We foresee greater interest in the subarea of metabolomics and sports nutrition, with an emphasis on using the results to design personalized, precision nutrition and recovery strategies for maximizing the effects of exercise-induced health benefits.

## Figures and Tables

**Figure 1 proteomes-10-00027-f001:**
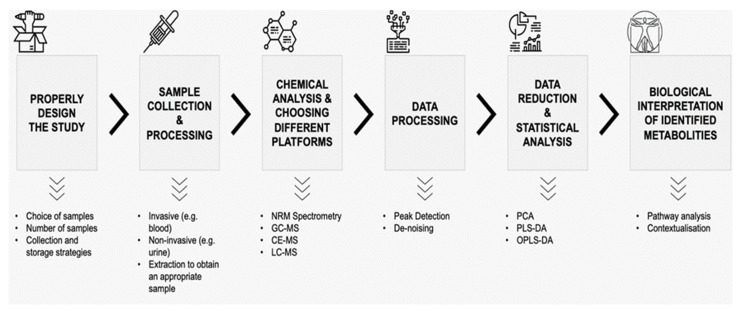
Workflow of a metabolomic study. NMR—nuclear magnetic resonance, GC-MS—gas chromatography–mass spectrometry, CE-MS—capillary electrophoresis–mass spectrometry, LC-MS—liquid chromatography–mass spectrometry, PCA—principal component analysis, PLS-DA—partial least squares discriminant analysis, OPLS-DA—orthogonal partial least squares discriminant analysis.

**Table 1 proteomes-10-00027-t001:** Advantages and disadvantages of biological samples typically used in sportomics.

Type of Sample	Invasivity of Collection Method	Advantages	Disadvantages
Blood	Very invasive	Appropriate for all methods of analysis. Includes endogenous metabolites and contains all molecules secreted or excreted by different tissues.	It contains proteins and lipoproteins. It makes it difficult to identify small metabolites via NMR. Metabolic degradation of blood analytes with enzymes in the sample.
Tissue	Very invasive	Furnishes the most accurate indicator of local metabolites. Supplies high concentrations of detectable metabolites.	Limited amounts of samples can be taken. Often the concurrent presence of high molecular weight proteins.
Urine	Minimally invasive	Contains stable metabolites. Macromolecules are almost absent. Contains endogenous and exogenous compounds. Possibility to collect several samples. Simple storage and shipment.	The presence of a high concentration of salts and urea can be a problem in M.S. platforms. Can be contaminated by bacteria, new metabolites’ synthesis, and changes in the original metabolic profile. Diet and environmental conditions can significantly affect the the sample.
Saliva	Minimally invasive	Presence of low-molecular-weight molecules. Mirrors the physiological conditions of the body. Simple storage and shipment.	Contaminated by bacteria that can activate the new synthesis of metabolites. Presence of high-molecular-weight proteins. The composition of saliva can be affected by physiological and pathological conditions of the mouth. Lower concentrations of endogenous metabolites with respect to the blood.
Stool	Technically non-invasive	Sampling is possible regularly and in sufficient quantities. Contains a mixture of metabolites. Provides useful insight on metabolic status, health/disease state, and symbiosis with the gut microbiome.	Biological variance and significant variations in metabolites’ composition due to the different regions of the source of the sample. Diet and environmental conditions can significantly affect the complexity of the sample.

**Table 2 proteomes-10-00027-t002:** Summary of studies investigating the use of metabolomics analysis in elite sports team athletes.

References	Subjects	Collection	Type of BS	Metabolomics Analytical Techniques and Aims of the Study
Santone et al., 2014	n = 14 elite professional soccer players from the Italian Lega Pro team (C1)	Before and after the level 1 Yo-Yo intermittent recovery test	Saliva	^1^H-NMR. Determining exercise-induced metabolites changes
Ra et al., 2014	n = 122 male soccer players (intercollegiate athletes who belonged to a soccer team)	Vefore and after 3 consecutive days (90 min game per day) of a 3-match tournament	Saliva	CE-TOFMS. Identifing metabolites in fatigued players
Barton et al., 2017	n = 40 professional international male rugby union players and n = 46 controls	1 time point	Urine and feces	^1^H-NMR, R.P., and HILIC for urine. UPLC-MS and GC-MS-targeted SCFA for feces. Identifing differences between athletes and non-athletes
Al-Khelaifi et al., 2018	n = 116 elite athletes from different sports disciplines who participated in national or international sports events (n = 41 male rugby players, n = 8 volleyball players (4F/4M), n = 1 male baseball players, n = 4 male basketball players, n = 62 male soccer players)	Spare samples, collected by doping control	Serum	NTMBMS combined with UHPLC to metabolomics profiling of athletes from different team sports
Al-Khelaifi et al., 2018	n = 331 elite athletes from different sports (n = 315 male soccer players; n = 16 male rugby players participated in national or international sports events)	Spare samples, collected by doping control	Serum	NTMBMS combined with UHPLC to analyze the presence of various xenobiotics that potentially originate from nutritional supplements
Al-Khelaifi et al., 2019	n = 338 from different sports (n = 315 male soccer players, n = 16 male rugby players, n = 2 male baseball players, n = 1 volleyball player, n = 3 male basketball players, n = 1 female hockey player participated in national or international sports events)	Spare samples, collected by doping control	Serum	NTMBMS combined with UHPLC to compare metabolic differences in athletes with high versus low/moderate cardiovascular demand
Al-Khelaifi et al., 2019	n = 490 from different sports (n = 315 male soccer players, n = 16 male rugby players, n = 2 male baseball players, n = 1 male volleyball player, n = 3 male basketball players, n = 1 female hockey player participated in national or international sports events)	Spare samples, collected by doping control	Serum	NTMBMS combined with UHPLC to investigate genetically influenced metabolites that discriminate elite athletes from non-elite athletes and to identify those associated with endurance sports
Pitti et al., 2019	n = 17 female professional team soccer players from the Italian Res Roma	Before and after a *Coppa Italia* soccer match	Saliva	^1^H-NMR to assess metabolic changes in saliva metabolites occurring during a soccer match
Akazawa et al., 2019	n = 12 female volleyball players from the top level of Japanese college team	1 time point in the early morning after 12 h overnight fast	Saliva	CE-TOFMS to investigate the impact of QoS on metabolite levels
Pintus et al., 2020	n = 21 professional soccer players from the Italian First Division (Serie A)	3 time points 2nd, 6th, and 16th day of pre-season	Urine	^1^H-NMR to study exercise-induced metabolite changes during pre-season
O’Donovan et al., 2020	n = 37 international Irish athletes from 16 different sports, many of whom participated in the 2016 Summer Olympics (n = 10 field hockey players)	1 time point	Feces and urine	NMR and UPLC-MS analysis for fecal samples and NMR, GC-MS, and UPLC-MS analysis for urine. Exploring the impact of training load and type of exercise on metabolites
Khoramipour et al., 2020	n = 70 male basketball players from the top level of Iran national top-league	8 time points, before and after each quarter	Saliva	^1^H-NMR to investigate the salivary metabolic fluctuations between the four 10 min quarters of high-level basketball games
Quintas et al., 2020	n = 80 professional soccer players from FCB under 18-teams and 2 reserve teams as volunteers	5 time points, 1 in pre-season and 4 in-season	Urine	UPLC-MS to study the association between the external load and the urinary metabolome as a surrogate of the metabolic adaptation to training
Hudson et al., 2021	n = 7 male rugby players from an elite English Premiership squad	8 time points over a competitive week including gameday	Urine, blood, and saliva	NMR spectroscopy to investigate the urine, serum, and saliva metabolic changes over a competitive week including gameday
Al-Muraikhy et al. 2021	n = 126 young elite male soccer players who participated in national or international sports events	Spare samples, collected by doping control	Serum	Waters ACQUITY -UPLC and Thermo Scientific Q-Exactive high resolution/accurate mass spectrometer interfaced with heated electrospray ionization (HESI-II) to study the metabolic alterations and identify the metabolic predictors of leukocyte telomere length (LTL)
Marinho et al., 2022	n = 23 male soccer players from a Brazilian elite championship team (Serie A)	3 time points over a 2 soccer matches interspersed by 72 h of recovery	Urine	^1^H-NMR and subsequent PCA and OPLS-DA to study metabolic changes immediately post a first match, the day after (20 h after), and after (20 h post) a second match

BS: biological sample; QoS: quality of sleep. CE-TOFMS: capillary electrophoresis and time-to-flight mass spectrometry; NTMBMS: non-targeted metabolomics-based mass spectroscopy; UHPLC: ultra-high-performance liquid chromatography; ^1^H-NMR: protonic untargeted metabolomics; UPLC: ultra-performance liquid chromatography.

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
