# Peer review of "Metabolomics in Team-Sport Athletes: Current Knowledge, Challenges, and Future Perspectives"

_proteomes, 2022, doi:10.3390/proteomes10030027_

Round 1

Reviewer 1 Report

In their review titled "Metabolomics in team-sport athletes: current knowledge, challenges, and future perspectives", the authors discuss the application of the metabolomics approach for investigating the exercise-induced metabolites changes, identifying metabolites related to fatigue,  and other applications in sport. The review is divided into several parts, like used methods, sample collection techniques, data analysis, case studies, and discussion. The review is written interestingly, it is informative and easy to read.

However, I have a few minor remarks and questions

Minor remarks:

Table 1. 

Missing space in “identification of small metabolites by NMR.Metabolic”

Table 2. Ref. Barton et al. 2017:

 Extra space in “players, n= 46 controls” 

Table 2. Refs. Al-Khelaifiet al. 2018 and Al-Khelaifiet al. 2019:

It is not clear why the total amount of subjects is more than the sum of groups.

For example:

n=478 elite athletes from different sports [n=315 male soccer players; n=16 male rugby players participated in national or international sports events]

315+16 = 331, not 478.

Please, clarify.

Questions:

The main question is about regarding some inconsistencies in the conclusions about the use of urine and saliva as noninvasive samples in the future. However, in the text and Table 1, you say that the urine could contain a high concentration of salts, contaminated by bacteria, influenced by diet and environment. And saliva is contaminated with bacteria and has endogenous metabolites with low concentrations.

Also from general considerations, metabolites in stool and urine are not only the result of the current biochemical reactions but also the result of food digestion. And the microbiota in the gut will significantly affect the composition of the metabolites.

Therefore, blood remains the most appropriate object to study the metabolism of ongoing biochemical reactions. And in sports, it is the body's reaction to various activities, i.e., changes that occur quite quickly. For example, the heart rate, immediately after exercise is increased, and then it returns to normal quite quickly.

Of course, blood sampling is an invasive method. And it would be better to get away from it.

There is such a variant of low-invasive blood sampling as the use of dried blood spots, in which blood is taken from a puncture of the finger on special paper. This both makes it easier to store the samples and allows you to easily get rid of blood proteins and red blood cells. Of course, this type of analysis requires highly sensitive methods of analysis, but it is possible to take a larger blood spot for analysis, thereby increasing the amount of the substance.

Actually, the second question - why don't you consider dried blood spots as a possible sample type in further research?

Nevertheless, the review is very interesting, and I recommend it for publication after minor revision.

Author Response

In their review titled "Metabolomics in team-sport athletes: current knowledge, challenges, and future perspectives", the authors discuss the application of the metabolomics approach for investigating the exercise-induced metabolites changes, identifying metabolites related to fatigue,  and other applications in sport. The review is divided into several parts, like used methods, sample collection techniques, data analysis, case studies, and discussion. The review is written interestingly, it is informative and easy to read.

We thank the reviewer for the comments that will greatly improve the quality of our manuscript

However, I have a few minor remarks and questions

Minor remarks:

Table 1. 

Missing space in “identification of small metabolites by NMR.Metabolic”

Response: Thank you for this correction. We added the space after NMR.

Table 2. Ref. Barton et al. 2017:

 Extra space in “players, n= 46 controls” 

Response: We added “and” after the word players.

Table 2. Refs. Al-Khelaifiet al. 2018 and Al-Khelaifiet al. 2019:

It is not clear why the total amount of subjects is more than the sum of groups.

For example:

n=478 elite athletes from different sports [n=315 male soccer players; n=16 male rugby players participated in national or international sports events]

315+16 = 331, not 478.

Please, clarify.

Response: Thank you for this important comment. In these studies authors enrolled athletes from different sports: individual and team sports. We have corrected the total number of participants as you suggested. It was a typo.

Questions:

The main question is about regarding some inconsistencies in the conclusions about the use of urine and saliva as noninvasive samples in the future. However, in the text and Table 1, you say that the urine could contain a high concentration of salts, contaminated by bacteria, influenced by diet and environment. And saliva is contaminated with bacteria and has endogenous metabolites with low concentrations. Also from general considerations, metabolites in stool and urine are not only the result of the current biochemical reactions but also the result of food digestion. And the microbiota in the gut will significantly affect the composition of the metabolites.

Therefore, blood remains the most appropriate object to study the metabolism of ongoing biochemical reactions. And in sports, it is the body's reaction to various activities, i.e., changes that occur quite quickly. For example, the heart rate, immediately after exercise is increased, and then it returns to normal quite quickly. Of course, blood sampling is an invasive method. And it would be better to get away from it.

Response: We agree with these considerations. As you stated, every sample presents pros and cons.

There is such a variant of low-invasive blood sampling as the use of dried blood spots, in which blood is taken from a puncture of the finger on special paper. This both makes it easier to store the samples and allows you to easily get rid of blood proteins and red blood cells. Of course, this type of analysis requires highly sensitive methods of analysis, but it is possible to take a larger blood spot for analysis, thereby increasing the amount of the substance. Actually, the second question - why don't you consider dried blood spots as a possible sample type in further research?

Response: We agree with you that dried blood spot (DBS) technique minimizes invasiveness and reduces storage and shipping costs and could be another potential strategies to study metabolites. Indeed, the WADA announced the use of DBS for the 2022 Beijing Winter Olympic Games Paralympic Games in routine doping control. However, in our literature review we didn’t find any use in the sports team setting, but a potential use in the future it is possible. We added this statement in the conclusion and future perspectives paragraph.

Nevertheless, the review is very interesting, and I recommend it for publication after minor revision.

Response: thank you for appreciating our work.

Reviewer 2 Report

The work is very interesting in terms of its subject matter. It seems to me that she was treated a little too general. The problems faced by analysts when interpreting the results of the metabolite analyzes have not been fully described - e.g.
1) what affects the obtained results is the type of detector used, advantages and disadvantages of detectors, e.g. TOF, QUAD, and a description of the advantages and disadvantages of MRM in spectrometry masses, 2) description of the advantages and disadvantages of the methods used in the purification of biological samples, not only the advantages and disadvantages of biological samples typically used in sportomics,
3) the statistics mentioned by the authors (PCA, PLS, etc.) contain traps that may falsify the results, it would be worth writing down in a few sentences what to avoid when using these specific statistic techniques.

Author Response

The work is very interesting in terms of its subject matter. It seems to me that she was treated a little too general.

Response: thank you for appreciating our efforts. As suggested, we improved the discussion regarding the interpretation of the results of the metabolite analyzes.

The problems faced by analysts when interpreting the results of the metabolite analyzes have not been fully described - e.g. 
1) what affects the obtained results is the type of detector used, advantages and disadvantages of detectors, e.g. TOF, QUAD, and a description of the advantages and disadvantages of MRM in spectrometry masses,

Response: We agree, that the obtained results are affected by the type of detector used. We expanded the discussion of the advantages and disadvantages of the detector, e.g. TOF and we described the advantages and disadvantages of MRM in spectrometry masses.

2) description of the advantages and disadvantages of the methods used in the purification of biological samples, not only the advantages and disadvantages of biological samples typically used in sportomics, 

Response: Thanks for this advice to improve the description of the advantages and disadvantages of the methods used in the purification of biological samples. We expanded the discussion in paragraph 3.

3) the statistics mentioned by the authors (PCA, PLS, etc.) contain traps that may falsify the results, it would be worth writing down in a few sentences what to avoid when using these specific statistic techniques.

Response: We agree. We wrote in a few sentences what to avoid when using these specific statistic techniques.

Reviewer 3 Report

My comments 

In this manuscript, a comprehensive review of all current articles on exercise metabolomics is presented in a well-written manner. However, several points should be clarified in the manuscript before publication.

1. The authors mention that the molecular mass threshold (Line 34) for metabolomes is 1.5 kDa. In the absence of an official definition, I suggest citing other articles with different cutoff points to provide a range rather than a threshold.

2. The authors describe the human species as possessing of 110,000 metabolites and 46,000 metabolic pathways (lines 51-52). There is a lack of clarity in the description of this definition. It is important to note that the amount of metabolites and pathways that are essential for cell survival is much smaller than these numbers. In addition to essential metabolites, the 110,000 metabolites also include other compounds derived from food, drugs, and microbes. For more accurate definitions, and updated numbers of compounds and pathways, the authors should cite the most recent literature (Nucleic acids research vol. 50,D1 (2022): D622- D631). 

Ref.: Wishart, David S et al. "HMDB 5.0: the Human Metabolome Database for 2022." Nucleic acids research vol. 50,D1 (2022): D622- D631. D631. doi:10.1093/nar/gkab1062

3. As different body fluids are produced by different organs, simultaneous analyses of multiple samples of various body fluids (saliva, blood, urine, etc.) may produce inconsistent results. This point should be mentioned by the authors.

4. Some information is lacking in the description "liquid chromatography-mass spectrometry, can detect a broader range of polar and non-polar metabolites..." (line 92). In LC-MS, polar and non-polar metabolites are separated using HILIC and RP columns, respectively. The GC-MS, on the other hand, is capable of separating both polar and non-polar metabolites through the use of one type of column. This concept should be included in the manuscript by the authors.

5. In the manuscript, the authors discuss two approaches: targeted metabolomics and untargeted metabolomics (lines 121-125). Moreover, the authors mention that many published exercise metabolomics studies lack quantification of metabolites (line 389). However, the authors did not link the two descriptions adequately. It is possible to obtain accurate quantitation of target metabolites with triple quadrupole MS, whereas the data obtained from untargeted metabolites can be only semi-quantitative with high-resolution MS. To better link their description, the authors should add this information. 

Author Response

In this manuscript, a comprehensive review of all current articles on exercise metabolomics is presented in a well-written manner. However, several points should be clarified in the manuscript before publication.

 We thank the reviewer for the comments that will greatly improve the quality of our manuscript

1.The authors mention that the molecular mass threshold (Line 34) for metabolomes is 1.5 kDa. In the absence of an official definition, I suggest citing other articles with different cutoff points to provide a range rather than a threshold.

Response: Thank you for this comment. We added a suggested range.

2.The authors describe the human species as possessing of 110,000 metabolites and 46,000 metabolic pathways (lines 51-52). There is a lack of clarity in the description of this definition. It is important to note that the amount of metabolites and pathways that are essential for cell survival is much smaller than these numbers. In addition to essential metabolites, the 110,000 metabolites also include other compounds derived from food, drugs, and microbes. For more accurate definitions, and updated numbers of compounds and pathways, the authors should cite the most recent literature (Nucleic acids research vol. 50,D1 (2022): D622- D631). 

 Ref.: Wishart, David S et al. "HMDB 5.0: the Human Metabolome Database for 2022." Nucleic acids research vol. 50,D1 (2022): D622- D631. D631. doi:10.1093/nar/gkab1062

 Response: Thank you for this comment. We updated and cited the most recent literature (Wishart et al., 2022)

3.As different body fluids are produced by different organs, simultaneous analyses of multiple samples of various body fluids (saliva, blood, urine, etc.) may produce inconsistent results. This point should be mentioned by the authors.

 Response: Thank you for this comment. We mentioned this point in the paragraph's conclusion and future perspectives.

4. Some information is lacking in the description "liquid chromatography-mass spectrometry, can detect a broader range of polar and non-polar metabolites..." (line 92). In LC-MS, polar and non-polar metabolites are separated using HILIC and RP columns, respectively. The GC-MS, on the other hand, is capable of separating both polar and non-polar metabolites through the use of one type of column. This concept should be included in the manuscript by the authors.

Response: Thank you for this comment. We included the suggested concept.

5. In the manuscript, the authors discuss two approaches: targeted metabolomics and untargeted metabolomics (lines 121-125). Moreover, the authors mention that many published exercise metabolomics studies lack quantification of metabolites (line 389). However, the authors did not link the two descriptions adequately. It is possible to obtain accurate quantitation of target metabolites with triple quadrupole MS, whereas the data obtained from untargeted metabolites can be only semi-quantitative with high-resolution MS. To better link their description, the authors should add this information. 

Response: Thank you for this comment and for suggesting linking the two descriptions of targeted and untargeted approaches. We added the statement that triple quadrupole MS can help the quantification of target metabolites, and that untargeted metabolomics can offer a semi-quantification of compounds.

Round 2

Reviewer 2 Report

Accept in present form 

Author Response

We would like to thank the reviewer for the constructive comments and for approving the publication of the manuscript